# Virtual Surgical Planning in Orthognathic Surgery: Two Software Platforms Compared

Pasquale Piombino [1,*], Vincenzo Abbate [1], Lorenzo Sani [1], Stefania Troise [1], Umberto Committeri [1], Emanuele Carraturo [1], Fabio Maglitto [1], Giacomo De Riu [2], Luigi Angelo Vaira [2] and Luigi Califano [1]

1    Maxillofacial Surgery Unit, Department of Neurosciences, Reproductive and Odontostomatological Sciences, University of Naples "Federico II", 80138 Naples, Italy
2    Maxillofacial Surgery Operative Unit, University Hospital of Sassari, Viale San Pietro 43/B, 07100 Sassari, Italy
*    Correspondence: piombino@unina.it; Tel.: +39-081-746-2087

**Featured Application: Orthognatic Surgery.**

**Abstract:** Over 70% of patients suffering from dentofacial deformities mention esthetics as the biggest issue pushing them to look for orthodontic and orthognathic treatment. At present, several pieces of software for computer-aided surgery have been released on the market. This surgical planning software allows surgeons to manipulate digital representations of hard and soft tissue profile tracings and subsequently morph the pretreatment image to produce a treatment simulation. The aims of this study were to investigate and find the difference between two of the most used pieces of digital software in pre-surgical planning for patients affected by dentofacial deformities by using the following parameters: usability, validity, timing, accessibility, efficacy, and predictability of the pre-surgical planning. Analyzing the results obtained from our study, it is correct to define both software tools useful and valid in digital surgical planning for the treatment of patients with dentofacial deformities. Each software has negligible differences in performance that do not in any way affect the success of surgical planning. The IPS software represents a valid alternative to the most popular and tested Dolphin Imaging software, and we are even inclined to evaluate it as better in terms of accuracy, effectiveness, and reliability.

**Keywords:** dentofacial deformities; orthognathic surgery; comparative study; computer-aided surgery; virtual planning; 3D surface scanning

## 1. Introduction

Over 70% of patients suffering from dentofacial deformities mention esthetics as the biggest issue pushing them to look for orthodontic and orthognathic treatment [1–5]. At the beginning of orthognathic surgery in 1970s, surgeons used patient photographs with profile tracings. In the 1980s, computer-generated line drawings of the profile based on hard tissue changes became possible, and, by the mid-1990s, treatment simulation software started to allow surgeons to virtually plan surgeries [6–8]. Technological evolution has led to the production of computers with increasingly greater computing capabilities. This factor has made it possible to develop the CAD CAM and virtual design technology upon which virtual planning software is now based. At present, several software tools for computer-aided surgery are available on the market: Dentofacial Planner Plus (Dentofacial Software, Toronto, ON, Canada) (DFP), IPS (IPSCaseDesigner KLS Martin Group), Quick Ceph (Quick Ceph Systems, San Diego, CA, USA), and Dolphin Imaging (Dolphin Imaging Software, Canoga Park, CA, USA) (DI), among others [9–14].

The aim of each of these software tools is always to realize surgical splints that can allow the surgeon to transfer the virtual planning to the operative field. Each software has different graphic interfaces and different levels of automation and customization.

This surgical planning software allows surgeons to manipulate digital representations of hard and soft tissue profile tracings and subsequently morph the pretreatment image to produce a treatment simulation. To the best of our knowledge, there are currently no studies that analytically evaluate the intuitiveness, profitability, and speed of digital workflows of the schedules among the various software tools available.

Therefore, the goal of this study is to investigate and find the difference in terms of usability between one of the most used digital software tools for virtual planning in patients affected by dentofacial deformities, the well-proven Dolphin Imaging, and the emerging IPS Case Designer.

## 2. Materials and Methods

A retrospective, observational, non-profit study was set up on a sample of patients admitted to our maxillofacial unit for dentofacial deformities from 2019 to 2020. We randomly enrolled 10 patients selected from our medical records. All clinical investigations and procedures were conducted according to the principles expressed in the Declaration of Helsinki. Ethical approval to access and use the data was obtained from the Federico II Research Ethic Committee (371/2019). The pre-operative (Cone Beam Computer Tomography-CBCT) (Planmeca ProMax® 3D s, PLANMECA OY, ASENTAJANKATU 6, FIN-00880 HELSINKI, FINLAND) scans, as well as facial scans and dental cast 3D models (PLANMECA®, https://www.planmeca.com/it/informazioni-stampa/press-room/planmeca-emerald-s (accessed on 14 September 2022) (STL [standard triangulation language] format), were acquired within one month of surgery. In order to better standardize the comparison, all of the Cone Beam CT scans were acquired by the same operator (E.C.), and the patients were scanned in the centric relation (the most retruded, unstrained position of the mandibular condyle within the temporomandibular joint (TMJ), that is, within the glenoid fossa) [15]. The facial scans were acquired by the same operator keeping the patient's head position in the NHP by using 3DMD system (3dMD Technologies Ltd., 3200 Cobb Galleria Parkway, #203, Atlanta, GA 30339, USA). The two medical software tools used for comparison were: Dolphin Imaging (version 11.9, Dolphin Imaging Software, Canoga Park, CA, USA) and IPS (IPSCaseDesigner KLS Martin Group, version 2.0). To reduce the errors in the study, the data were processed by the same computer (Dell Precision Tower 5820, ©2021 Dell Inc., Aliso Viejo, CA, USA) and acquired by the same operator for both groups. We based our comparison on seven different outcomes to better highlight the differences between the two pieces of software.

### 2.1. Time

We used time as a variable to underline the software performances, measuring all of the phases of acquisition of the 3D scans, facial scans, intraoral scans (we took this measurement indirectly by scanning the casts), the data loading (Cone Beam CT processing), and the time required for complete surgical planning. We also calculated the "learning curve", calculating the time required for complete surgical planning for all 10 patients by an inexperienced operator, to visualize the reduction in the time required for planning at increasing levels of experience of the operator.

### 2.2. Fundamental Investigations

Fundamental investigations refer to the specific elements (instrumental investigations) that the software needs to upload in the system to complete a surgical planning.

### 2.3. Linearity of Planning Path

We used this tool to underline differences between the two software tools in terms of simplicity and intuitiveness of use of the software itself. To pursue a standardized comparison, we selected as a comparative item the number of "windows" opened to proceed in the planning path without any form of interruption or trouble caused by the software.

### 2.4. Surgical Transfer

Surgical transfer is meant to define which of the two software tools allows the surgeon to transfer more accurately what has been planned in the operating room by providing occlusal splints and planning reports. Surgical transference was evaluated through a numerical evaluation in tenths carried out by the operators (2 surgeons and 4 residents) about the level of appreciation after facing with the effectiveness of the surgical planning and the accuracy of occlusal splints in the operating room. Participants evaluated the model using survey ratings based on a 5-point Likert scale (Figure 1).

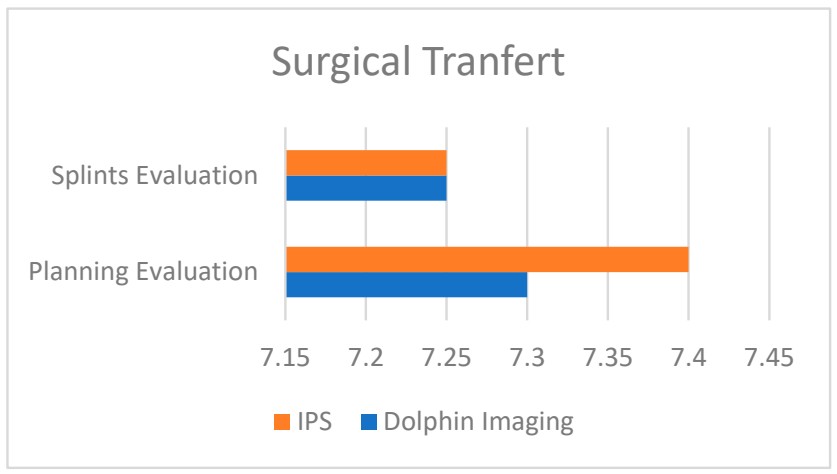

**Figure 1.** Evaluation of the surgical transfer.

### 2.5. Basal Specifics and

We included in the comparison information about the basal specifics, such as minimal requirements.

### 2.6. Statistical Analysis

Data management and analysis were performed using a *t*-test (Microsoft Excel Statistics for Windows, Version 3.0., IBM Corp.: Armonk, NY, USA) based on the Student's *t*-test for independent samples. The level of significance was set at 0.05 ($p < 0.05$).

## 3. Results

### 3.1. Sample Features

Seven males and three females aged from 18 to 35 years of age with an average age of $24 \pm 3.4$ y affected by dentofacial deformities (7 patients affected by class III malocclusion, 3 patients affected by class II malocclusion).

The results of the comparison are listed below.

### 3.2. Time

The two software tools, Dolphin Imaging and IPS, have comparable complete **planning times**. Small differences show the IPS software to be, with the same data of a single patient, slightly faster, with an average time saving of 18.4 ($\pm$19.2) minutes.

The learning curves show a **slope**, although similar, that is higher in the points inherent to the first measurements with the Dolphin Imaging software, suggesting also in this case a greater difficulty (and, therefore, a longer planning time) for the first measurements of a novice operator.

**CBCT acquisitions** show a substantial difference between the two software tools of /2.06 $\pm$ 0.52/ seconds in favor of Dolphin Imaging.

Analyzing the data acquisition time records of the **intraoral scans** of the two software tools, it appears that Dolphin Imaging is faster than IPS in the 10 measurements by /1.03 $\pm$ 0.24/ seconds.

The analysis of the results related to the acquisition of **3D facial scans** once again attests how the Dolphin Imaging software proves to be more performing, guaranteeing an average time saving of /1.46 ± 0.46/ seconds (Table 1).

**Table 1.** Software comparison data: planning time, acquisition time for CBCT, intraoral scan, facial scan.

| Patients | Time to Complete Planning | | | Acquisition Time of CBCT | | | Acquisition Time of Oral Scans | | | Acquisition Time of Facial Scans | | |
|---|---|---|---|---|---|---|---|---|---|---|---|---|
| | Dolphin Imaging | IPS | Variation | Dolphin Imaging | IPS | Variation | Dolphin Imaging | IPS | Variation | Dolphin Imaging | IPS | Variation |
| Patient 1 | 231 m | 164 m | 67 m | 12.77 s | 14.21 s | 1.44 s | 4.20 s | 5.41 s | 1.21 s | 5.56 s | 7.02 s | 1.46 s |
| Patient 2 | 116 m | 95 m | 21 m | 12.67 s | 15.01 s | 2.34 s | 4.10 s | 5.20 s | 1.10 s | 5.47 s | 6.49 s | 1.02 s |
| Patient 3 | 97 m | 66 m | 31 m | 12.98 s | 14.69 s | 1.71 s | 4.23 s | 4.96 s | 0.73 s | 5.21 s | 6.78 s | 1.57 s |
| Patient 4 | 55 m | 52 m | 3 m | 12.15 s | 14.10 s | 1.95 s | 4.57 s | 5.17 s | 0.60 s | 5.67 s | 7.59 s | 1.92 s |
| Patient 5 | 58 m | 49 m | 9 m | 12.49 s | 14.43 s | 1.94 s | 4.13 s | 5.19 s | 1.06 s | 5.51 s | 7.14 s | 1.63 s |
| Patient 6 | 50 m | 46 m | 4 m | 12.53 s | 14.73 s | 2.20 s | 3.87 s | 4.87 s | 1.00 s | 5.13 s | 6.86 s | 1.73 s |
| Patient 7 | 57 m | 50 m | 7 m | 11.90 s | 15.08 s | 3.18 s | 4.27 s | 5.27 s | 1.00 s | 5.41 s | 7.17 s | 1.76 s |
| Patient 8 | 49 m | 41 m | 8 m | 12.70 s | 14.20 s | 1.50 s | 3.97 s | 4.99 s | 1.02 s | 5.87 s | 6.93 s | 1.06 s |
| Patient 9 | 52 m | 33 m | 19 m | 12.79 s | 14.63 s | 1.84 s | 4.21 s | 5.71 s | 1.50 s | 5.76 s | 6.74 s | 0.98 s |
| Patient 10 | 44 m | 29 m | 15 m | 12.69 s | 15.19 s | 2.50 s | 4.25 s | 5.35 s | 1.10 s | 5.49 s | 7.18 s | 1.69 s |
| Average time | 80.9(±57.7) min | 62.5(±40.2) min | 18.4(±19.2) min | 12.57 ± 0.32 s | 14.63 ± 0.39 s | 2.06 ± 0.52 s | 4.18 ± 0.19 s | 5.21 ± 0.26 s | 1.03 ± 0.24 s | 5.51 ± 0.23 s | 6.99 ± 0.30 s | 1.46 ± 0.46 s |

### 3.3. Fundamental Investigations

Both software tools require 3D facial scans (not necessary for both pieces of software but very useful to obtain a realistic planning), facial CBTC, and intraoral scans. Both software tools do not require the execution and download of any additional radiographic data (orthopantomographic X-ray and tele-skull X-ray in the antero-posterior and latero-lateral projections), but, although capable of deriving them from CBTC data, only Dolphin Imaging uses these instrumental investigations in programming (Table 2).

**Table 2.** Fundamental investigation required for a complete surgical planning.

| *Dolphin Imaging* | *IPS* |
|---|---|
| Cone Beam CT (minimum request) | Cone Beam CT (minimum request) |
| 3D Facial Scans | 3D Facial Scans |
| Oral Scans (minimum request) | Oral Scans (minimum request) |
| Rx Orthopantomographic (optional) | Rx Orthopantomographic (not requested) |
| Rx Telecranium AP-LL (optional) | Rx Telecranium AP-LL (not requested) |
| Occlusal Scans (not requested) | Occlusal Scans (optional) |

### 3.4. Linearity of Programming Path

The number of windows needed by the operator to complete a plan with the Dolphin Imaging software is 17, three more units compared to the windows needed to complete a plan with the IPS software (14). Dolphin Imaging software uses the first three windows for loading the patient data sheet, unlike the IPS software (2). IPS software requires one less window to ensure the occlusion overview, integrated in the next window for the construction of the occlusion scan, a missing step in the programming of the Dolphin Imaging software (programming does not require these data).

### 3.5. Surgical Transfer

The effectiveness of surgical planning appears to be comparable between the Dolphin Imaging and IPS software tools, with a slight preference for the latter software, showing an average evaluation value of 7.4 against the average value of 7.3 of the Dolphin Imaging software. A differential of 0.1 is, on the other hand, lower than the value of the SD (0.75), making the differential itself statistically irrelevant. The analysis of the results referring to the evaluation of occlusal splints in the operating room shows an absolute comparability, showing an average value of 7.25 for both pieces of software (Figure 1).

### 3.6. Basal Specific

**Minimal Requirement** (directly taken from https://dolphinimaging.it/it/products/3d/dolphin-3d/; https://www.klsmartin.com/en/products/individual-patient-solutions/ips-casedesigner/?L=%2525252Fetc%2525252Fpasswd (accessed on 14 September 2022)) (Table 3).

**Table 3.** Minimal requirement to work with the two software.

| Recommended (Dolphin Imaging) | Minimum (Dolphin Imaging) | Component | Minimum (IPS) | Recommended (IPS) |
|---|---|---|---|---|
| Windows 7 SP1, Windows 8, Windows 8.1, or Windows 10 | Windows 7 SP1, 32 or 64 bit | **Operating System** | Windows 7, 8, 10, 64 bit | Mac OS Yosemite or higher |
| 4 GB or higher | 2 GB (4 GB for Dolphin 3D) | **RAM** | 8 GB | 16 GB |
| Intel® Core™ i7 | Intel® Core™2 Duo | **CPU** | 2 core processor | i7 processor with high clock rate |
| 3D only: 512 MB video card, based on one of these DirectX 9.0 graphics engines or better: NVIDIA 8400 GS, ATI HD 5450, or Intel HD 4600 | Standard (not mentioned) | **Graphics Card** | Optimal 3D support (Open GL) | Min 1 GB own onboard memory, 2 GB for 4K/Retina displays |
| 1920 × 1080 | 1280 × 1024 at 24-bit color | **Screen Resolution** | 1440 × 900 | Full HD (1920 × 1080), 1920 × 1200 |
| 500 GB or higher | 3 GB free disk space only to install | **Disk Space** | 5 GB free disk space or more | |
| Broadband connection of 5 Mbps or faster | Broadband connection | **Internet** | Internet connection | Broadband Internet connection |

**Statistical Analysis** Data referring to planning measurements for the two software tools were judged significant with $p < 0.01$. Statistical analysis of the acquisition of oral scans, facial scans, and CBCT showed how the measurements were not statistically significant given the small sample size and the low variability of the data, with $p > 0.5$.

## 4. Discussion

Every year, thousands of patients elect to undergo combined surgical-orthodontic treatment to ease correction of severe jaw deformities [1–5]. Due to the complex nature of the dentofacial anatomy, orthognathic surgery often requires extensive pre-surgical planning [6–8]. In recent years, the rapid development of fast and affordable digital computers has revolutionized medical surgery [1]. This revolution has affected orthodontics in many ways, and the best improvement was achieved using surgical planning tools. Over the years, the advent of imaging software programs has proved to be useful for diagnosis, treatment planning, and outcome measurement [16]. Therefore, over the years, the use of virtual surgery planning software for education, pre-operative assessment, pre-surgical planning, and measurement has become very prevalent as, since recently, the market has offered various options for the surgeon to choose: Dentofacial Planner Plus (Dentofacial Software, Toronto, ON, Canada) (DFP), IPS (IPSCaseDesigner KLS Martin Group), Quick Ceph (Quick Ceph Systems, San Diego, CA, USA), and Dolphin Imaging (Dolphin Imaging Software, Canoga Park, CA, USA) (DI), among others.

The aim of our study was to make a comparison between two surgical planning software tools on the market for the study and planning of orthognathic surgery: the well-known and proven Dolphin Imaging and the emerging IPS. Willinger et al. discussed a comparison of these two VPS software tools based on accuracy of soft tissue prediction in two patients undergoing an intraoral quadrangular Le Fort II osteotomy [17] and a comparison based on feasibility, time consumption, and costs in a standardized workflow

for a modified intraoral quadrangular Le Fort II osteotomy (IQLFIIO) [18]. As we know, in the current literature, there is no work that compares the VPS Dolphin Imaging and the IPS discussing items such as accuracy, validity, time, and usability in the surgical planning of dentoskeletal dysmorphia.

Analysis of the results from the measurements of the acquisition times: analyzing the measurements, the first difference in performance is evident regarding the acquisition times of the CBCT data, with the average difference in acquisition in favor of the Dolphin software, showing more effective imaging in both single acquisition and long-run. Moreover, it is interesting to note how the data acquisitions of the 10 CBCTs are much more heterogeneous in the case of the IPS software, a sign of greater variability. Analyzing the data acquisition time records of the intraoral scans of both the Dolphin Imaging and IPS software tools, the first result is the variation in the acquisition times that shows the Dolphin Imaging software to be more performing, that is, faster than IPS in the 10 measurements.

Both software tools have comparable data homogeneity, demonstrated both in the 10 measurements of the two individual software tools and in the comparison between the two software acquisition times, specifying that this homogeneity can also be correlated to the relative "lightness" of the intraoral scan data. The analysis of the results related to the acquisition of 3D facial scans offers interesting insights, confirming once again how the Dolphin Imaging software proves to be more performing. Thus, it must be underlined that the 3D facial scans are, for both software tools, very useful but not strictly mandatory, as both can accomplish a complete planning without this tool. The analysis of the acquisition times highlights a relative heterogeneity of the IPS software alone, both in the single 10 measurements and in the comparison with the recordings of the Dolphin Imaging software, which, also in this case, proves to be faster. It would, therefore, seem that the Dolphin Imaging software is more powerful in managing a large amount of data.

As statistical analysis showed, the variances in the acquisition times (CBCT acquisition, oral scan acquisition, facial scas acquisition) are not statistically relevant ($p > 0.5$). This is probably due to the small sample (10 patients) at our disposal.

Analysis of the results related to the study of the linearity of the programming path: the analysis of the results related to the number of open windows in order to determine the linearity of the planning path demonstrates similarity.

The number of windows required by the operator to complete a planning with the Dolphin Imaging software is 17 windows, greater than the 14 windows required to complete a planning with the IPS software. In addition to the number of windows, the intrinsic difference in the specific purposes was also evaluated: the Dolphin Imaging software uses the first three windows to load the patient data sheet, unlike the two used by the IPS software.

The IPS software requires one less window to ensure the occlusion overview, integrated in the next window to load the occlusion scan, a missing step in the planning of the Dolphin Imaging software. Larger "saving" for the IPS software is given by the latter not needing to use the radiographic data (orthopantomography and telecranium).

The phases inherent to the actual planning can be absolutely overwritten for the two software tools.

Each software requires eight windows to plan osteotomies and design the occlusal splints. The difference in the number of open windows in planning would suggest a greater ease of use of the IPS software. Ultimately, the investigation into the linearity of the planning path does not show the predominance of one software over the other, bringing out overlapping results in terms of performance and usability.

Analysis of the results of the accuracy of the surgical transfer: the analysis of the results shows, according to the evaluations carried out by the operators, that the effectiveness of the surgical planning appears to be comparable between the two software tools, Dolphin Imaging and IPS, with a slight preference towards the latter software, showing an average evaluation value of 7.4 against the average value of 7.3 attributed to the Dolphin Imaging software. A difference of 0.1 is less than the value of the SD (0.75), making the differential

itself statistically irrelevant. The analysis of the results referring to the evaluation of occlusal splints in the operating room shows an absolute comparability, attributing to each software an average value of 7.25.

Analysis of the results regarding the fundamental investigations: both software tools require 3D facial scans, CBCT, and intraoral scans.. Both software tools do not require the execution and download of radiographic data (Rx orthopantomography and Rx telecranium in the antero-posterior and latero-lateral projections), which, for IPS, are not necessary for planning, while for Dolphin Imaging they can be extrapolated directly from the CBCT data.

Analysis of the results regarding the basal specifics: the basic specifications are comparable, with the only major difference being that the Dolphin Imaging software does not have a version available for the Mac OS operating system.

## 5. Conclusions

Analyzing the results obtained from our study, it is correct to define both software tools useful and valid in digital surgical planning for the treatment of patients with dentofacial deformities. Each software has negligible differences in performance that do not in any way affect the success of the surgical planning. In terms of the basic specifications, the IPS software represents a valid alternative to the most popular and tested Dolphin Imaging software in terms of accuracy, effectiveness, and reliability.

We can, therefore, conclude that the introduction of 3D virtual planning in orthognathic surgery has made it possible to significantly improve the phases preceding surgery by offering the surgeon the possibility of obtaining an accurate plan and reliability superior to that extrapolated from the single articulator. However, it should be noted that all of this cannot be separated from a careful clinical examination or from the preparation and experience of the surgeon.

**Author Contributions:** Conceptualization, P.P. and V.A.; methodology, U.C.; software, E.C.; validation, P.P., V.A. and F.M.; formal analysis, S.T. and L.S.; investigation, E.C.; resources, P.P.; data curation, E.C.; writing—original draft preparation, E.C.; writing—review and editing, L.A.V.; visualization, G.D.R.; supervision, L.C.; project administration, L.C.; funding acquisition, none. All authors have read and agreed to the published version of the manuscript.

**Funding:** This research received no external funding.

**Institutional Review Board Statement:** The study was conducted in accordance with the Declaration of Helsinki and approved by the Ethics Committee of Federico II Research Ethic Committee (371/2019 of 21 February 2020).

**Informed Consent Statement:** Informed consent was obtained from all subjects involved in the study. Written informed consent has been obtained from the patient(s) to publish this paper.

**Data Availability Statement:** Not applicable.

**Conflicts of Interest:** The authors declare no conflict of interest.

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
