# Peer review of "Virtual Surgical Planning in Orthognathic Surgery: Two Software Platforms Compared"

_applsci, doi:10.3390/app12189364_

Round 1

Reviewer 1 Report

The authors compared the differences between two commercially available virtual planning software, DI and IPS, for orthognathic surgery by several parameters: usability, validity, timing, accessibility, efficacy and predictability of the pre-surgical planning and the software costs. They concluded that both software tools are useful and valid in surgical planning for the treatment of patients with dentofacial deformities.

In this manuscript, there are several scientific concerns arisen from the reviewer.

Firstly, motivation and aim of this manuscript lack of research significance. It sounds more likely an evaluation report in describing the details of how the two same functional software in different interfaces use.

We would not say the selection of two of the most used digital orthognathic planning software, since the new available software, IPS, may not distributed worldwide yet.

Several flaws appear in the manuscript, such as, sign of number, unit missing in tables, non-English spelling and mis-spelling (Cone Beam Tc, etc), position of table captain.

In table I, it is meaningless to compare the upload time of the same processing files from the two software. What is the meaning of the special symbol ‘/’ using in the variation which can be a positive or a negative number. What we care is whether the planning outcomes from the software processing can meet the requirements of end users. The research study should be in the position of un-bias evaluation the goal of achievements but not the trivial.

In the section of surgical transfer at line 163. Please explain how the evaluation value of the software effectiveness comes from. Please explain how to analysis the evaluation values of occlusal splints in operation room. Will it be possible a subjective feeling from different surgeons? Please explain.

Costs of software may be changed according to marketing strategy in different countries. It is not appropriate to compare the list price in a research paper.

Window numbers doesn’t mean performance or efficacy of the software.

Duplicate references appear at [3] and [15], [9] and [10].

Base on the knowledge of reviewer, soft tissue prediction was one of the functions of Dolphin Imaging software. A suggestion from the reviewer, to compare the prediction and surgery outcomes in different stages of the soft tissue may be an interesting topic to the readers.

Author Response

Dear Reviewer,

Thank you for all your suggestions and comments.

For the corrections, please see the attachment.

Reviewer 2 Report

Dear authors,

the paper appears well structured in terms of methodology. It is very interesting to ring alternative softwares to plan Orthognatic surgery.

the results are clearly exposed

further plannings are needed in future to prove more significantly your thesis

Author Response

Dear reviewer,

Thank you very much for your support. We aim to publish a bigger paper in which we discuss openly about the post-operative outcomes of both software, providing a larger number of plannings in order to make the cohort and therefore the study more significant.

Reviewer 3 Report

This is an interesting commercial study about two software for virtual surgical planning. The authors would like to make a comparison between two surgical planning software on the market in terms of accuracy, validity, time and usability. It is well-designed. However, I have questions for the evaluation of the surgical transfer:

1.      How do you evaluate the occlusal splints ? Have you test the splint fitness in the patient`s mouth before surgery? And How ?

2.      Please explain more about the effectiveness of surgical planning. What is the average evaluation value of 7.4 ?

      3 The authors stated that IPS software is even inclined to evaluate it better in                terms of accuracy, effectiveness and reliability. However, a successful surgery            not only depends on an accurate virtual planning but also depends on the                  surgeon if he or she could loyal to this planning from any kind of simulation                software

Author Response

Dear reviewer,

Thank you very much for your support and instructions. I am going to provide explanations to your comments and suggestiond point per point right down:

1) We evaluated accuracy of occlusal splints superimposing the scans of the splints with their .STL files. We did not mention any splints fitness comparison as we would like to describe it more precisely in a bigger study about the post-operative accuracy outcomes of IPS and DI that is about to be ended.

2)Surgical transference was evaluated through a numerical evaluation in tenths carried out by the operators (2 Surgeons and 4 residents) about the level of appreciation after facing with the effectiveness of the surgical planning and the accuracy of occlusal splints in the operating room. Participants evaluated the model using survey ratings based on a 5-point Likert scale. This considerations have been inserted into the revisioned manuscript in the materials and methods section.

3)In the same section, we stated how the surgeries were performed by the same 2 surgeons, in order to eliminate any possible human variability about the fidelity in following the sofware programming path.

Reviewer 4 Report

Abstract : line 48 point behind market instead of ,

Introduction: line 81 allows instead of allow

Materials an methods:

Sorry, in my opinion you have to describe the statistical analysis in the section.

Results:

I would recommend to write the sentence in line 119 without brackets. 

Line 125 : the two softwares, dolphin imaging and ips, have….

Line 126: how did measure the difference between the programs? Did you mean only the time in which you made the movements or did you mean also the whole preparation time prior to the movements ( loading of ct-data, preparation of the splits etc). For comparison: in our department 20 min are more or less the time which we need for one patient if we look only at the movements. A difference of difference of 18.4 min seems not plausible for me….

Line 132: do you really think, that a difference of two seconds would be a substantial difference between the softwares if you describe a difference of 18.4 min as slightly? I cannot follow…

Table 1:

There are no units written in the table. I guess that the overall-time has minutes as unit an the acquisition time has seconds???

I think that you have to declare if you did a real intraoral scan for the intraoral situation or if you scanned the casts. Further i think, that a difference of the acquisition time is of less interest. The time of matching the cast- or intraoral scans with the facial scan and the ct-scan ist the most important value…

Line 144:

I think, that you have to declare, the programs do not need additional radiographic data. But the ct-scan is also radiographic data and in the view of exposure to radiation the only examination which really does matter

Table 2: I have never used dolphin but i have a lot experience with ips: I think that you have to declare that the ct-scan and a scan of the teeth are the minimum requirement for ips. You can add a therapeutic occlusion scanning the casts (occlusion scan). We do our simulations without a surface scan. Thus, the face scan (surface) is not necessary for ips and think in dolphin it would be similar.

Line 161: I wonder why dolphin obviously does not ask for an occlusion scan. Does the program only offer an automatic occlusion?

Line 166: Sorry, but i do not understand the evaluation value. How was this value calculated? I cannot find the declaration in the materials and methods.

Figure 1: surgical transfer

The text does not describe how the evaluation of the splints and the planning was done. I also cannot comprehend, how the values were calculated.

Table 3: ok, but is this part of a scientific article?

Line 192:

Sorry, but the explanation of the statistical analysis is part of materials and methods and not of the result section. I is hard to comprehend, for what you exactly have used the t-test. For the acquisition-times? Which t-test was used and were the prerequisites (normal distribution) for a t-test given? The level of significance has also to be declared in the methods. The possible reasons, why no significance could be observed should be part of the discussion section.

Discussion:

Line 242:

Since the simulation can be performed excellently without a face scan in IPS, you can hardly notice a performance difference between the programs due to this feature.

Line 250-266:

I think, that i must not necessary an advantage to have less windows to get the end of the process. Please consider: The more windows you have, the more features you might choose in the program. What about the presentation of the N.alveolaris inferior. IPS offers a perfect feature to show the Position of Alveolar Nerve. Does dolphin offer this, too?

Line 280:

That is not true. You can choose the program function to set the therapeutic occlusion in ips. 

Conclusion:

Sorry, thr conclusion is too long and not supported by your results.

Overall:

Digital planning has recently replaced the former manual planning more and more. Thus, studies are needed to check the accuracy and performance of digital planning software. Analyses to check the accuracy of the surgical transfer should, in my view, include a comparison of the planning with the surgical result, which would be possible, for example, by means of a comparison between planned and achieved position of the jaws and the overlying soft tissues. In my opinion, the time for loading the data is not important because these activities can easily be delegated or handed over to KLS-Martin in case of IPS. The planning time itself depends from my point of view more on the complexity of the case than on the technology. With a case number of only 10 in the present study it is hard to make a valid statement here

Author Response

Dear Reviewer, 

thank you cery much for all your comments and suggestions;

In order to take a look for all responses, please see the attachment

Round 2

Reviewer 1 Report

For a research paper, motivation and aim in this manuscript was meaningless. It sounds more likely a report of comparing two OGS planning software in their similar function but different interfaces use. Commercial software interface can be changed and upgraded depend on hardware platform upgrading. Time of software function applied in different level platform can be varied. The higher performance platform use, the faster the function achieved. If the time comparision not conduct in the same hardware platform, the comparision outcome will be bias.
Some of reviewer suggestions were put off for the reason of next publication.
Tables and figures were in incorrect format and pool orgnization.
Spelling check was still missing in both tables and figures.
It also needs an approval letter from IRB committee to access  patients' computer tomography.

Reviewer 4 Report

Dear authors, dear editor,

I am sorry, but I still maintain my initial assessment. The article could be improved significantly, but in my opinion, it is still not a real scientific manuscript. The measured times in a total of 10 cases are, in my view, completely irrelevant to the quality of the program. Already in my first assessment I had written that a difference in loading time of 2 seconds is clinically and practically not relevant at all. Dysgnathia cases can be very simple and easy to handle, but they can also be difficult and the subject of long discussions, so that with the total number of 10 it should be difficult to prove a real difference between the programs. In general, the performance of the programs should not be measured by time, but by the quality of the results. This was done only on the basis of personal evaluations with a simple scale of 10, although hard variables such as a comparison between planned and achieved position would be possible. I have also shown the work to my colleagues and they come to the same assessment as Reviewer 1 and myself.